# MATLABER: MATERIAL-AWARE TEXT-TO-3D VIA LATENT BRDF AUTO-ENCODER

## ABSTRACT

Based on powerful text-to-image diffusion models, text-to-3D generation has made significant progress in generating compelling geometry and appearance. However, existing methods still struggle to recover high-fidelity object materials, either only considering Lambertian reflectance, or failing to disentangle BRDF materials from the environment lights. In this work, we propose Material-Aware Text-to-3D via LAtent BRDF auto-EncodeR (**MATLABER**) that leverages a novel latent BRDF auto-encoder for material generation. The BRDF auto-encoder is trained with large-scale real-world BRDF collections, serving as a useful prior to constrain the generated material in a natural distribution. To further disentangle material from environment lights, we adopt a semantic-aware material regularization that motivates object parts with the same semantics to share similar materials. Through exhaustive experiments, our approach demonstrates the superiority over existing ones in generating realistic and coherent object materials. Moreover, high-quality materials naturally enable multiple downstream tasks such as relighting and material editing.

## 1 INTRODUCTION

3D asset creation is imperative for various industrial applications such as gaming, film, and AR/VR. Traditional 3D asset creation pipeline involves multiple labor-intensive and time-consuming stages (Labschütz et al., 2011), all of which rely on specialized knowledge and professional aesthetic training. Thanks to the recent development of generative models, recent text-to-3D pipelines that automatically generate 3D assets from purely textual descriptions have received growing attention, due to their rapid advances in generation quality and efficiency, as well as their potential of significantly reducing the time and skill requirement of traditional 3D asset creation.

Gradually optimizing the target 3D asset represented as NeRF (Mildenhall et al., 2020) or DMTET (Shen et al., 2021) through the SDS loss (Poole et al., 2023), compelling geometry and appearance can be obtained by these text-to-3D pipelines (Jain et al., 2022; Mohammad Khalid et al., 2022; Poole et al., 2023; Chen et al., 2023; Lin et al., 2023; Wang et al., 2023). However, as shown in Figure1, they still struggle to recover high-fidelity object materials, which significantly limits their real-world applications such as relighting under novel illuminations. Although attempts have been made to model Lambertian reflectance (Poole et al., 2023; Lin et al., 2023) and bidirectional reflectance distribution function (BRDF) (Chen et al., 2023), in their designs, the neural network responsible for predicting materials has no sufficient motivation and clues to unveil an appropriate material that obeys the natural distribution, especially under fixed light conditions, where their predicted material is often entangled with environment lights.

In this work, we resort to existing rich material data to learn a novel text-to-3D pipeline that effectively disentangles material from environment lights. In fact, despite the lack of paired datasets of material and text descriptions, there exist large-scale BRDF material datasets such as MERL BRDF (Matusik et al., 2003), Adobe Substance3D materials (Adobe), and the real-world BRDF collections TwoShotBRDF (Boss et al., 2020). Therefore, we propose Material-Aware Text-to-3D via LAtent BRDF auto-EncodeR (**MATLABER**) that leverages a novel latent BRDF auto-encoder to synthesize natural and realistic materials that accurately align with given text prompts. The latent BRDF auto-encoder is trained to embed a data-driven BRDF prior to its smooth latent space so that MATLABER can predict BRDF latent codes instead of BRDF values, which regularizes the

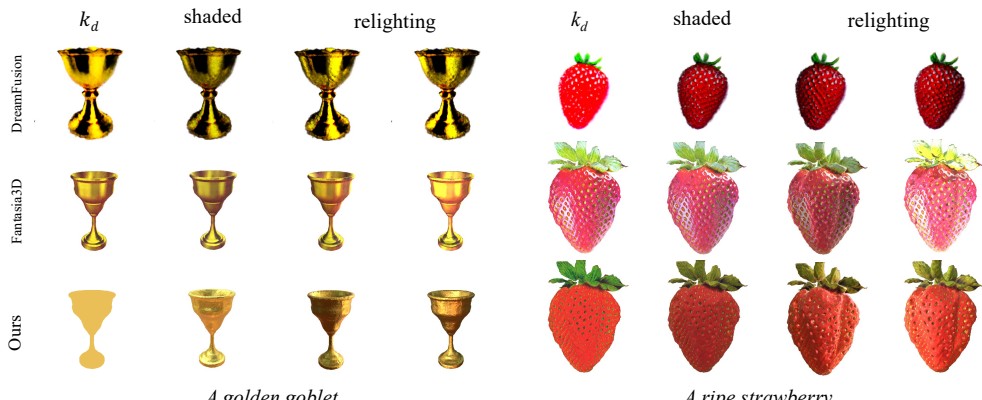

Figure 1: Text-to-3D generation aims to synthesize high-quality 3D assets aligning with given text descriptions. Despite the impressive appearance, representative methods like DreamFusion (Poole et al., 2023) and Fantasia3D (Chen et al., 2023) still fail to recover high-fidelity object materials. Specifically, DreamFusion only considers diffuse materials while Fantasia3D always predicts BRDF materials entangled with environment lights. Based on a latent BRDF auto-encoder, our approach is capable of generating natural materials for 3D assets, enabling realistic renderings under different illuminations.

predicted BRDF to fall in the natural BRDF distribution. Moreover, inspired by the fact that object parts with the same semantics (*e.g.* the head or the handle of a hammer) tend to share the same material, we employ a material smoothness regularizer that penalizes large deviations of material values from the average of each semantic part of the object. This regularizer enforces smoother diffuse materials and further encourages disentangling materials from environment lights.

Thanks to the latent BRDF auto-encoder and material smoothness regularizer, our approach ensures the realism and coherence of object materials, achieving ideal disentanglement of geometry and appearance. As shown in Figure 1, our approach can create 3D assets with high-fidelity material, outperforming previous state-of-the-art text-to-3D pipelines. More importantly, the effective estimation of object materials naturally allows various operations such as relighting and material editing, which can hardly be achieved before our work. It's noteworthy that these downstream tasks are extremely crucial for a series of real-world applications, taking a solid step towards more applicable 3D content creation.

## 2 RELATED WORK

### 2.1 TEXT-TO-IMAGE GENERATION

In recent years, we have witnessed significant progress in text-to-image generation empowered by diffusion models. By training on large-scale text-image paired datasets, diffusion models can implicitly link semantic concepts and corresponding text prompts, and thus is capable of generating various and complex images of objects and scenes (Nichol et al., 2021; Saharia et al., 2022; Ramesh et al., 2022; Rombach et al., 2022). While GLIDE (Nichol et al., 2021) obtains text embeddings via a pretrained CLIP (Radford et al., 2021) model, Imagen (Saharia et al., 2022) and eDiff-I (Balaji et al., 2022) adopt larger language models such as T5 (Raffel et al., 2020) to achieve more diverse image synthesis and a deeper level of language understanding. To enable text-to-image training on limited computational resources, Stable Diffusion (Rombach et al., 2022) leverages the latent diffusion model (LDM) and trains its diffusion model on the latent space instead of the pixel space, demonstrating highly competitive performance in terms of quality and flexibility. However, all of these works are constrained to the 2D domain and ignore the huge demand in the 3D field.

### 2.2 TEXT-TO-3D GENERATION WITH 2D SUPERVISION

With the great success of text-to-image synthesis, text-to-3D generation has gained significant attention in recent years as a promising way to generate high-quality 3D content from simple textual

input. Different from text-to-image generation, no massive text-3D paired data is available, making it infeasible to train a text-to-3D model as in training a text-to-image model. Alternatively, DreamField (Jain et al., 2022) and CLIP-mesh (Mohammad Khalid et al., 2022) explore the 2D supervision by using a pre-trained CLIP model to optimize the underlying 3D representations, such as NeRFs and meshes, to achieve high text-image alignment scores for multi-view renderings. Later, DreamFusion (Poole et al., 2023) utilizes, for the first time, a powerful 2D text-to-image diffusion model (Saharia et al., 2022) as a prior and introduces an effective Score Distillation Sampling (SDS) loss to guide the optimization. Based on the promising SDS loss, Magic3D (Lin et al., 2023) optimizes 3D objects in two consecutive stages and further improves the rendered resolution of generated 3D objects from $64$ to $512$, showing impressive performance in text-to-3D generation. In Fantasia3D (Chen et al., 2023), Chen *et al* proposes to represent 3D objects with flexible DMTET representation (Shen et al., 2021) and model the appearance via BRDF modeling, which can successfully generate compelling geometry and photorealistic object appearance. Recently, Prolific-Dreamer (Wang et al., 2023) proposes a more advanced guiding loss, Variational Score Distillation (VSD), for more diverse and high-fidelity object generation. Unlike these works, we aim to recover the inherent material information in text-to-3D generation, empowering more downstream applications such as relighting and material editing.

## 2.3 Material Estimation

In the community of computer vision and graphics, researchers aim to estimate surface materials for decades. The Bidirectional Reflection Distribution function (BRDF) is the most widely used material model, which characterizes how a surface reflects lighting from an incident direction toward an outgoing direction (Nicodemus, 1965; Li et al., 2018). One line of research works (Lensch et al., 2001; 2003; Gao et al., 2019; Bi et al., 2020) targets recovering the reflectance from known 3D geometry while some methods (Aittala et al., 2013; 2015; Thanikachalam et al., 2017; Hui et al., 2017; Kang et al., 2018; Deschaintre et al., 2018; Guo et al., 2020) only focus on BRDF acquisition of 2D planar geometry. Moreover, the simultaneous acquisition of 3D geometry and materials has also gained a surge of interest and several solid papers (Nam et al., 2018; Barron & Malik, 2014; Boss et al., 2021b; Zhang et al., 2021; Munkberg et al., 2022) have already achieved compelling results on in-the-wild scenarios. Specifically, Neural-PIL (Boss et al., 2021b) and NeRFactor (Zhang et al., 2021) both leverage BRDF datasets for material priors. Unlike these methods, we aim to create high-quality surface materials aligning with the provided text prompts. Concurrent to this work, Fantasia3D (Chen et al., 2023) also explores the automatic creation of surface materials and object geometries from given language models. However, their obtained materials are unluckily entangled with lights and thus intractable to put into new environments.

## 3 Method

We present Material-Aware Text-to-3D via LAtent BRDF auto-EncodeR (**MATLABER**), aiming for photorealistic and relightable text-to-3D object generation. In the following, in Section 3.1 we at first briefly introduce the text-to-3D pipeline adopted in this work. Then in Section 3.2, we review appearance modeling in prior works and analyze their intrinsic deficiencies in the relighting scenarios. Finally, we introduce the latent BRDF auto-encoder in Section 3.3 and discuss how to incorporate it into material-aware text-to-3D generation in Section 3.4. An overview of our framework is illustrated in Figure 2.

## 3.1 Score Distillation Sampling (SDS)

Here we introduce the basic text-to-3D pipeline (Poole et al., 2023) which achieves high-quality text-to-3D generation with a pre-trained text-to-image diffusion model. It represents the scene with a modified Mip-NeRF (Barron et al., 2021), which can produce an image $x = g(\theta)$ at the desired camera pose. Here, $g$ is a differentiable renderer, and $\theta$ is a coordinate-based MLP representing the target 3D content. To obtain the optimal $\theta$ given the input text description, this pipeline adopts a progressive optimization process, which tries to maximize the similarity between the text description and images rendered from the target 3D content. Following common practice, we use the Score

Distillation Sampling (SDS) loss to compute the similarity and update $\theta$:

$$\nabla_\theta \mathcal{L}_{\text{SDS}}(\phi, x) = \mathbb{E}_{t,\epsilon}\left[w(t)(\epsilon_\phi(x_t; y, t) - \epsilon)\frac{\partial x}{\partial \theta}\right], \tag{1}$$

where $w(t)$ is a weighting function. And $\phi$ is the pre-trained text-to-image diffusion model, which has the learned denoising function $\epsilon_\phi(x_t; y, t)$, with $\epsilon$, $x_t$, $t$ and $y$ being the sampled noise, noisy image, noise level, and text embeddding respectively. To ensure reproducibility, we follow (Chen et al., 2023) and leverage the publicly available latent diffusion model (LDM) like Stable Diffusion (Rombach et al., 2022) as the pre-trained text-to-image diffusion model.

## 3.2 APPEARANCE MODELING IN TEXT-TO-3D GENERATION

To employ the SDS loss, prior works adopt different schemes to model the appearance of 3D objects. DreamFusion (Poole et al., 2023) leverages a reflectance model similar to (Bi et al., 2020; Pan et al., 2021; Boss et al., 2021a; Srinivasan et al., 2021) and only considers diffuse reflectance (Lambert, 1760; Ramamoorthi & Hanrahan, 2001) while rendering multi-view images. Although generating various 3D objects with appealing appearances, DreamFusion fails to model specular reflectance, which is an indispensable term in the material that leads to photorealistic renderings under different illuminations. Magic3D (Lin et al., 2023) uses more advanced DMTET (Shen et al., 2021) as the scene representation and targets at higher-fidelity 3D models but still adopts Lambertian shading when texturing the object meshes extracted from DMTET. Hence, their synthesized 3D objects all lack complete material information, significantly limiting their applications in real-world scenarios.

To achieve more photorealistic and relightable rendering, Fantasia3D (Chen et al., 2023) introduces the spatially varying Bidirectional Reflectance Distribution Function (BRDF) into text-to-3D generation, which represents the material with three components (McAuley et al., 2012), namely the diffuse term $\boldsymbol{k}_d \in \mathbb{R}^3$, the roughness and metallic term $\boldsymbol{k}_{rm} \in \mathbb{R}^2$ (*i.e.*, roughness $k_r$ and metalness factor $m$), as well as the normal variation term $\boldsymbol{k}_n \in \mathbb{R}^3$. According to the convention in (Karis & Games, 2013), the specular term $\boldsymbol{k}_s$ is computed with $\boldsymbol{k}_s = (1 - m) \cdot 0.04 + m \cdot \boldsymbol{k}_d$. For a specific surface point $\boldsymbol{x}$ with normal $\boldsymbol{n}$ and outgoing view direction $\boldsymbol{\omega}_o$, the final rendering $L(\boldsymbol{x}, \boldsymbol{\omega}_o)$ can be obtained following the rendering equation (Kajiya, 1986):

$$L(\boldsymbol{x}, \boldsymbol{\omega}_o) = \int_\Omega L_i(\boldsymbol{x}, \boldsymbol{\omega}_i) f(\boldsymbol{x}, \boldsymbol{\omega}_i, \boldsymbol{\omega}_o; \boldsymbol{k}_d, \boldsymbol{k}_s, k_r)(\boldsymbol{\omega}_i \cdot \boldsymbol{n})\mathrm{d}\boldsymbol{\omega}_i, \tag{2}$$

where incident light $L_i$ comes from the direction $\boldsymbol{\omega}_i$ and BRDF $f$ is related to the diffuse term $\boldsymbol{k}_d$ and the specular term $\boldsymbol{k}_s$. However, in Fantasia3D (Chen et al., 2023), the metalness factor $m$ is usually set to 0 and thus the specular term $\boldsymbol{k}_s$ is actually ignored during the appearance modeling. Moreover, they utilize a fixed HDR environment map with uniform brightness distribution as the environment lights all the time. Despite its appealing appearance under fixed illuminations, Fantasia3D always predicts materials entangled with environmental lights, which leads to unrealistic renderings under novel lighting conditions.

## 3.3 LATENT BRDF AUTO-ENCODER

Confronted with the limitations of prior works, we take both diffuse and specular terms into consideration for appearance modeling. Specifically, for a surface point $\boldsymbol{x}$, we aim to estimate the Cook-Torrence (Cook & Torrance, 1982) BRDF parameter $\boldsymbol{k} \in \mathbb{R}^7$ including diffuse $\boldsymbol{k}_d \in \mathbb{R}^3$, specular $\boldsymbol{k}_s \in \mathbb{R}^3$, and roughness $k_r \in \mathbb{R}$. However, directly optimizing on standard BRDF space in the text-to-3D pipeline lacks necessary constraints and may make the predicted materials fall into invalid BRDF regions. As mentioned in Section 1, we resort to data-driven BRDF priors that are learned from real-world BRDF data collections. Specifically, we propose to train a latent BRDF auto-encoder, turning BRDF prediction into BRDF selection by predicting a latent code of the auto-encoder. This alternative not only guarantees the validity and quality of decoded materials but also substantially eases the task complexity.

We train the latent BRDF auto-encoder on TwoShotBRDF (Boss et al., 2020). TwoShotBRDF is a real-world BRDF material dataset containing $11,250$ high-quality SVBRDF maps of size $768 \times 768$ collected from various online sources, where each pixel represents an independent BRDF parameter. Following (Berthelot et al., 2018; Boss et al., 2021b), interpolating auto-encoders are adopted here

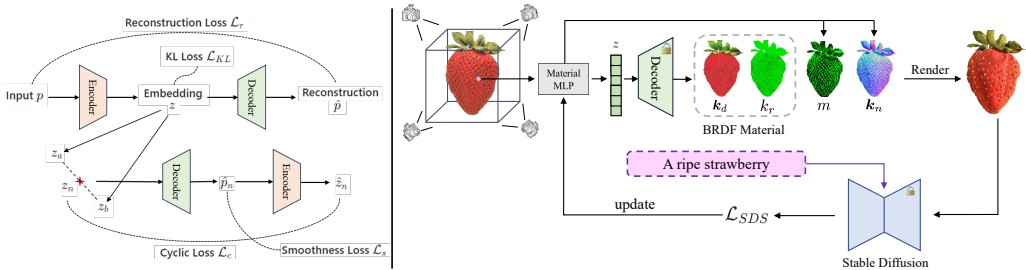

Figure 2: **Left**: Our latent BRDF auto-encoder is trained on the TwoShotBRDF dataset with four losses, *i.e.*, reconstruction loss, KL divergence loss, smoothness loss, and cyclic loss. Imposing KL divergence and smoothness loss on latent embeddings encourages a smooth latent space (Boss et al., 2021b). **Right**: Instead of predicting BRDF materials directly, we leverage a material MLP $\Gamma$ to generate latent BRDF code $z$ and then decode it to 7-dim BRDF parameters using our pretrained decoder, where only diffuse $k_d$ and roughness $k_r$ terms are utilized. The material network is also responsible for predicting metalness $m$ and normal variation term $k_n$. Similar to prior works, the SDS loss can be applied to the rendered images, which empowers the training of our material MLP network. (Note that, roughness $k_r$ and metalness $m$ are both scalars and we visualize them with the green channel here.)

for smoother interpolation in the latent space. As illustrated in Figure 2, our BRDF auto-encoder consists of an MLP encoder $\mathcal{E}$ and an MLP decoder $\mathcal{D}$. Given a BRDF parameter $k$, the encoder will generate a 4-dim latent code $z = \mathcal{E}(k)$, which is then reconstructed to a 7-dim BRDF $\hat{k} = \mathcal{D}(z)$ via the decoder network. Here, apart from the standard $L_2$ reconstruction loss $\mathcal{L}_r$, we additionally leverage a Kullback–Leibler (KL) divergence loss $\mathcal{L}_{\text{KL}} = KL(p(z)||\mathcal{N}(\mathbf{0}, \mathbf{I}))$ to encourage the smoothness of latent space. Moreover, for an ideal smooth latent space, the linearly interpolated embeddings $z_n = \alpha \cdot z_a + (1 - \alpha) \cdot z_b$ between two random latent codes $z_a$ and $z_b$ could be also decoded to a reasonable BRDF parameter $\hat{k}_n = \mathcal{D}(z_n)$. We, therefore, impose a smoothness loss $\mathcal{L}_s = \sum_n (\nabla_\alpha \mathcal{D}(z_n))^2$ for a batch of uniformly interpolated latent codes $z_n$ following (Boss et al., 2021b). After adding the $L_2$ cyclic loss $\mathcal{L}_c$ between interpolated code $z_n$ and the re-encoded counterpart $\hat{z}_n = \mathcal{E}(\hat{k}_n)$, the total loss for our latent BRDF auto-encoder training is a combination of four losses:

$$\mathcal{L} = \mathcal{L}_r + \lambda_{\text{KL}}\mathcal{L}_{\text{KL}} + \lambda_s\mathcal{L}_s + \lambda_c\mathcal{L}_c, \tag{3}$$

where $\lambda_{\text{KL}}$, $\lambda_s$ and $\lambda_c$ are all balancing coefficients. With this total loss function, our encoder and decoder networks are trained jointly.

### 3.4 MATERIAL-AWARE TEXT-TO-3D

In this work, we divide the whole text-to-3D generation into two consecutive stages: geometry generation and appearance generation.

**Geometry modeling.** Following Fantasia3D (Chen et al., 2023), we represent the 3D objects with the hybrid scene representation of DMTET owing to its superior performance in geometry modeling and photorealistic surface rendering. While generating the geometry, we employ the SDS loss on normal maps similar to strategies proposed in (Chen et al., 2023). For the DMTET predicted by a geometry MLP $\psi$, a differentiable render $g$ can render a normal map $n$ from a randomly sampled camera pose $c$ as follows: $n = g(\psi, c)$. Then, the SDS loss on normal maps $n$ will help the geometry MLP $\psi$ update until it converges to a satisfactory geometry aligning with the given text prompt.

**Appearance modeling.** Once obtaining the geometry of 3D objects, we can leverage our latent BRDF auto-encoder for appearance generation. As shown in Figure 2, for any point $x$ on the surface, we first apply the hash-grid positional encoding $\beta(\cdot)$ (Müller et al., 2022) and then use a material MLP $\Gamma$ parameterized as $\gamma$ to predict its BRDF latent code $z_x$, which is then transferred to 7-dim BRDF parameter $k_x$ via the pre-trained BRDF decoder $\mathcal{D}$ following:

$$z_x = \Gamma(\beta(x); \gamma), \quad k_x = \mathcal{D}(z_x). \tag{4}$$

Equipped with this crucial material information $k_z = [k_d, k_s, k_r]$, the point $x$ can be rendered with the aforementioned rendering Equation 2 under given incoming light $L_i(\omega_i)$ from direction

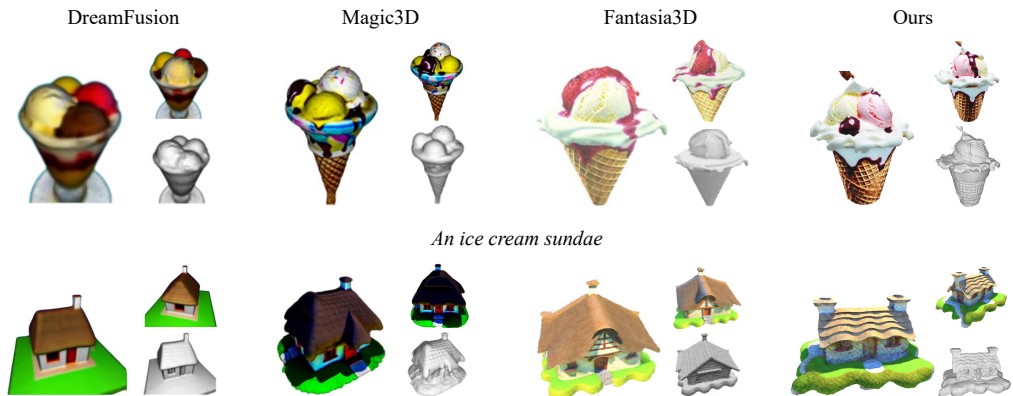

| DreamFusion | Magic3D | Fantasia3D | Ours |

*An ice cream sundae*

*A 3D model of an adorable cottage with a thatched roof*

Figure 3: Qualitative comparisons to baselines. Ours have more natural textures and richer details.

$\boldsymbol{\omega}_i$. However, the dataset TwoShotBRDF contains few metal materials, and such a data bias makes it extremely hard for our BRDF auto-encoder to recover metal materials during the appearance modeling. Hence, we only fetch the diffuse $\boldsymbol{k}_d$ and the roughness $k_r$ from the decoded 7-dim BRDF parameters $\boldsymbol{k_x}$. Moreover, the material network predicts an additional scalar value $m$ for computing the specular term $\boldsymbol{k}_s$ and a normal variation $\boldsymbol{k}_n$ term to compensate for the imperfect geometry following (Chen et al., 2023; Munkberg et al., 2022). Thus, the rendering results can be calculated using the split-sum method in (Karis & Games, 2013; Chen et al., 2023). To encourage the disentanglement between the predicted BRDF materials and environment lights, we randomly sample the environment maps from a pool of collections and randomly rotate the map during training.

**Semantic-aware material regularization.** Despite the powerful latent BRDF auto-encoder, the predicted diffuse sometimes still contains unacceptable highlights and shadows, *especially for the metal materials*. Based on the assumption that the same semantic part (e.g., the head or the handle of a hammer) tends to share similar materials, we further introduce a semantic-aware material regularization to reduce ambiguity. Specifically, for each semantic part of the generated object, we supervise the materials to move closer to the mean value of this part. Suppose the object includes $N$ parts $\{\mathbf{P}_i, \mathbf{P}_2, ..., \mathbf{P}_N\}$ represented with 3D masks and we punish the deviation with $L_2$-loss, this smooth regularization can be written as:

$$\mathcal{L}_{\text{mat}} = \sum_{i=1}^{N} \sum_{\boldsymbol{x} \in \mathbf{P}_i} \lambda_d |\boldsymbol{k}_d(\boldsymbol{x}) - \bar{\boldsymbol{k}}_d(\boldsymbol{x})|^2 + \lambda_r |k_r(\boldsymbol{x}) - \bar{k}_r(\boldsymbol{x})|^2 + \lambda_m |m(\boldsymbol{x}) - \bar{m}(\boldsymbol{x})|^2, \quad (5)$$

where $\bar{\boldsymbol{k}}.(\boldsymbol{x})$ denotes the average of material values in part $\mathbf{P}_i$, and $\lambda_d$, $\lambda_r$ and $\lambda_r$ are all loss weights. Note that, 3D masks here can be inferred with SAM (Kirillov et al., 2023) from multiple viewpoints after initial painting for the mesh, or obtained as a free lunch from blank models since in practice most of them naturally consist of several parts. On the task of text-to-3D, we empirically find this regularization is more effective than that proposed in prior works (Munkberg et al., 2022).

## 4 EXPERIMENTS

**Implementation details.** While training our latent BRDF auto-encoder, we set the batch size to 64, and loss balancing coefficients $\lambda_{\text{KL}}$, $\lambda_s$ and $\lambda_c$ to 2e-3, 1.0, and 1e-6, respectively. Our BRDF auto-encoder is trained for 30 epochs using the AdamW optimizer with a learning rate of 1e-4 on an NVIDIA A100 GPU. For the geometry modeling, we initialize the DMTET with either a 3D ellipsoid, a 3D cylinder, or a customized 3D model provided by users. While generating the materials, we set the material regularization weights $\lambda_d$, $\lambda_r$, and $\lambda_m$ to 1e8, 10, and 100. During the text-to-3D generation, the 3D object is optimized on 4 NVIDIA A100 GPUs with an AdamW optimizer, where each GPU loads 9 images rendered from randomly sampled camera poses. Specifically, our method spends $3,000$ iterations (learning rate 0.001) on geometry modeling and $2,000$ iterations (learning

Table 1: Mean opinion scores in range $1 \sim 5$, where 1 means the lowest score and 5 is the highest.

| Method | Alignment | Realism | Details | Disentanglement |
|---|---|---|---|---|
| DreamFusion | 3.97 ($\pm$ 0.66) | 3.56 ($\pm$ 0.43) | 3.23 ($\pm$ 0.61) | 3.48 ($\pm$ 0.59) |
| Magic3D | **4.01** ($\pm$ 0.59) | 3.84 ($\pm$ 0.72) | 3.70 ($\pm$ 0.66) | 3.14 ($\pm$ 0.89) |
| Fantasia3D | 3.76 ($\pm$ 0.82) | 4.17 ($\pm$ 0.65) | 4.27 ($\pm$ 0.75) | 2.93 ($\pm$ 0.95) |
| Ours | 3.81 ($\pm$ 0.75) | **4.35** ($\pm$ 0.60) | **4.31** ($\pm$ 0.70) | **3.89** ($\pm$ 0.65) |

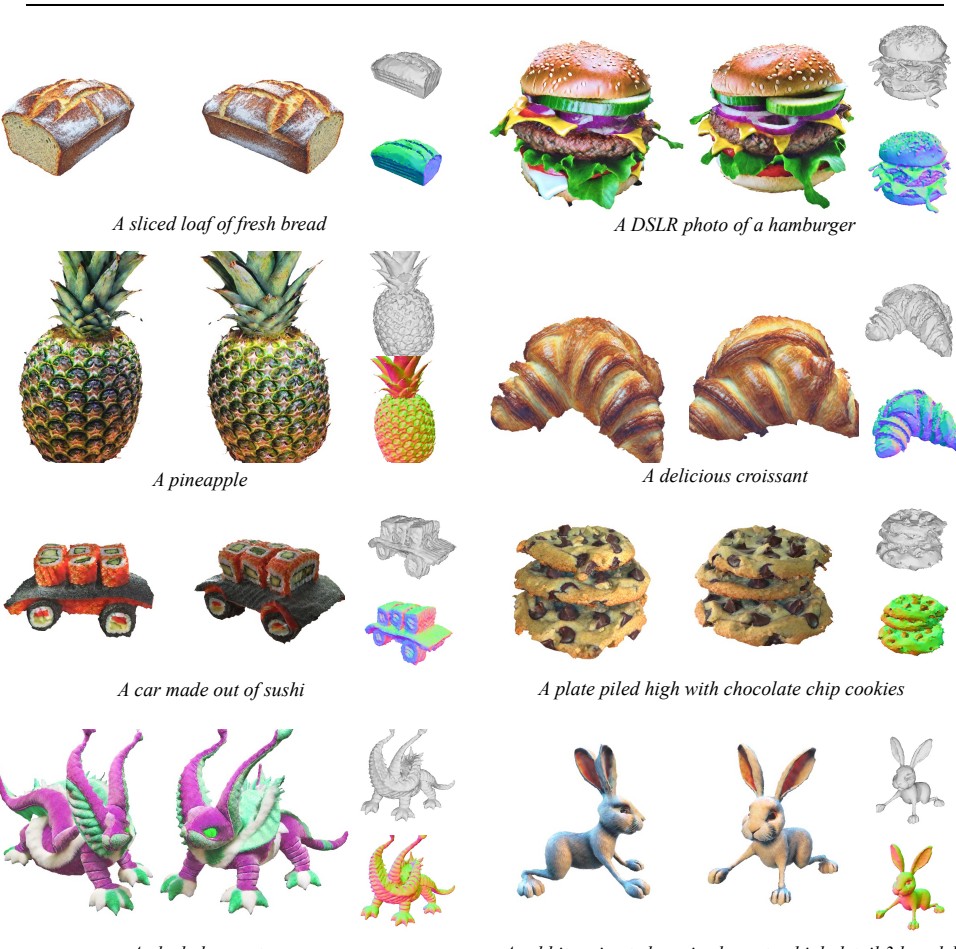

*A sliced loaf of fresh bread*   *A DSLR photo of a hamburger*

*A pineapple*   *A delicious croissant*

*A car made out of sushi*   *A plate piled high with chocolate chip cookies*

*A plush dragon toy*   *A rabbit, animated movie character, high detail 3d model*

Figure 4: The gallery of our text-to-3D generation results. Shapes, normal maps, and shaded images from two random viewpoints are presented here.

rate 0.01) on material generation. For the weight in score distillation sampling, we always adopt the *strategy I* proposed in Fantasia3D (Chen et al., 2023).

**Qualitative results.** Given text descriptions, our method MATLABER is capable of generating photorealistic 3D assets and simultaneously obtaining high-fidelity BRDF materials. In this section, we compare our approach with three representative methods, namely DreamFusion (Poole et al., 2023), Magic3D (Lin et al., 2023), and Fantasia3D (Chen et al., 2023). As shown in Figure 3, we present the comparative results given the same text descriptions. For Fantasia3D, 3D assets are synthesized with their official code, while results of DreamFusion and Magic3D are borrowed from their project pages owing to the inaccessibility of their model weights. Thanks to the latent BRDF auto-encoder, our method generates competitive geometry as Fantasia3D and demonstrates a better appearance with more natural textures and richer details. Moreover, the gallery of more 3D assets generated with MATLABER is provided in Figure 4.

**User study.** To evaluate the quality of generated 3D objects from human's perspective, we invited 80 volunteers to conduct a user study. For each participant, he/she will view 10 randomly selected

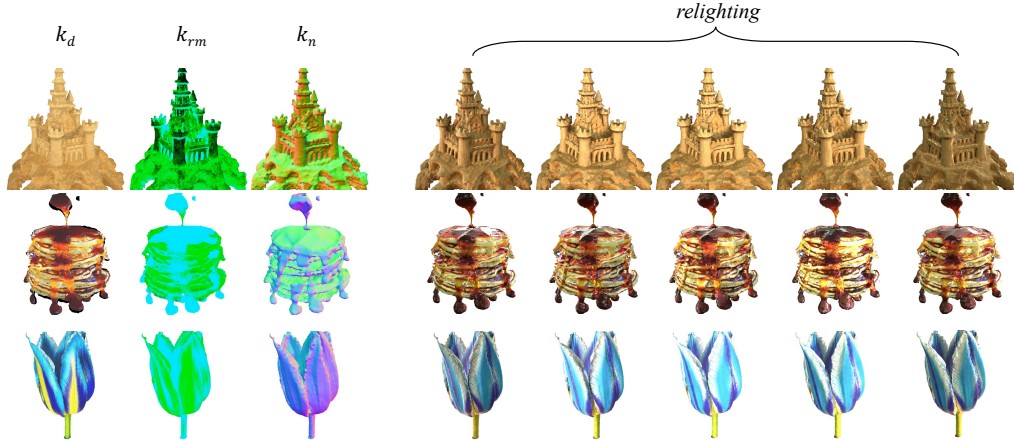

Figure 5: Relighting results. On the left side, we list the generated BRDF materials, including diffuse, roughness, metalness, and normal map. The relit images under a rotating environment light are presented on the right side.

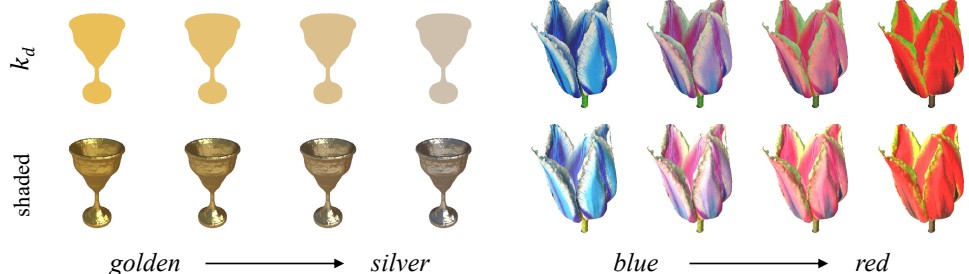

Figure 6: Material interpolation results. Thanks to the smooth latent space of our BRDF auto-encoder, we can conduct a linear interpolation on the BRDF embeddings. As can be observed here, a golden goblet will turn into a silver goblet while the color of the tulip changes from blue to red gradually.

3D objects generated by our approach and three baseline methods. They are asked to evaluate these 3D assets in four different dimensions: 'alignment', 'realism', 'details', and 'disentanglement'. Alignment means how closely the generated 3D objects match the given text prompts, and realism reflects the fidelity of synthesized mesh textures. Besides, users also need to judge the richness of details on these 3D objects. Finally, we present the corresponding diffuse materials and ask them whether the diffuse is disentangled from the environment lights.

In total, we collect 800 responses, and the comparative results are summarized in Table 1. Our method MATLABER achieves the best performance on three evaluation protocols, *i.e.*, realism, details, and disentanglement. Especially for disentanglement, ours outperforms three baseline methods by a large margin, showing the effectiveness of our proposed latent BRDF auto-encoder model. In terms of alignment, DreamFusion (Poole et al., 2023) and Magic3D (Lin et al., 2023) both capitalize on large-scale language models like T5 (Raffel et al., 2020) and hence demonstrate better alignment with the text descriptions. Unfortunately, these models are all not released for public usage.

**Relighting.** Since our model is capable of generating BRDF materials, we can implement relighting on these 3D assets. To validate the fidelity of our generated materials, we manually rotate an HDR environment map and thus obtain a series of HDR maps with different environment lights. As shown in Figure 5, the relit objects under different illuminations are all natural and photorealistic. In particular, for the maple syrup part in the pancake case, the diffuse contains almost no highlights and simultaneously the renderings under various illuminations demonstrate shiny results.

**Material interpolation and editing.** Thanks to the smooth latent space of our BRDF auto-encoder, we can also conduct material interpolation if given two different material descriptions. Figure 6 demonstrates a smooth material morphing on the goblet and tulip. For example, given the text

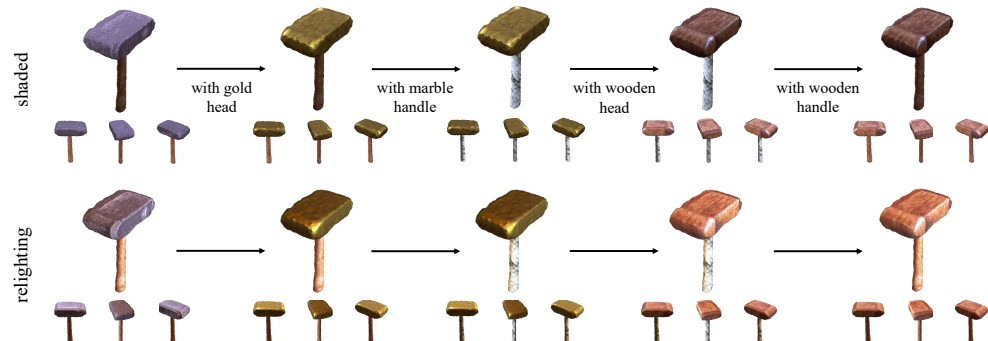

Figure 7: Material editing results. We start from the text prompt of "A hammer with iron head and wooden handle", and gradually edit this hammer to gold head, marble handle, wooden head, and wooden handle, respectively, under the guidance of corresponding text prompts, which demonstrates our model's capability of editing different parts.

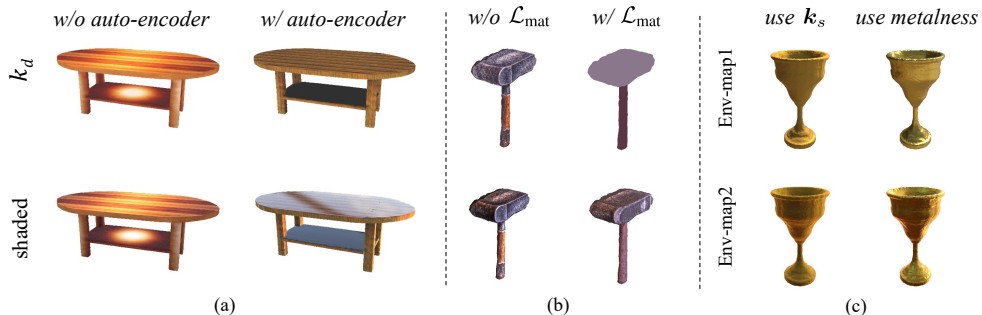

Figure 8: Ablation study. Figure (a) demonstrates our BRDF auto-encoder encourages the diffuse material to disentangle from the environment lights. In Figure (b), the material network is prone to bake some unsatisfactory highlights into the diffuse map if without the material smooth regularizer. In figure (c), we show that using metalness $m$ can achieve more shiny renderings on metal materials than that using the specular $\boldsymbol{k}_s$ from our decoder $\mathcal{D}$.

prompts of "A golden goblet" and "A silver goblet", we can obtain their corresponding material latent embeddings after the optimization, and the final results can be obtained via a naive interpolation on the latent embeddings. In Figure 7, we show our model is capable of flexibly editing different parts given the 3D masks and suitable text descriptions. As can be observed, we can modify the material of the head and handle alternately, across various materials of iron, wood, gold, and marble.

**Ablation study.** In Figure 8, several ablation studies are conducted to validate the effectiveness of our proposed modules or losses. As shown in Figure 8 (a) and (b), we show that the latent BRDF auto-encoder and material smooth regularizations boost disentanglement between the diffuse and illuminations. Figure 8 (c) illustrates our selection of metalness $m$ is clearly better than the predicted specular $\boldsymbol{k}_s$ on metal materials.

## 5 CONCLUSION

In this work, we propose MATLABER, a novel latent BRDF auto-encoder for material-aware text-to-3D generation. This auto-encoder is trained with large-scale real-world BRDF collections, providing implicit material prior for the appearance modeling in text-to-3D generation. Besides, we also introduce a material smoothness regularizer to encourage disentanglement between materials and illuminations. Thanks to the material prior and regularization, our approach can generate high-quality and coherent object materials in text-to-3D synthesis, achieving the ideal disentanglement of geometry and appearance. Moreover, the generated BRDF materials also support various operations such as relighting and material editing. Meanwhile, our latent BRDF auto-encoder potentially can be used in other tasks to predict materials instead of RGB values to enable the ability of relighting and material editing.

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
