# Supplemental Material for
# MATLABER: Material-Aware Text-to-3D via LAtent BRDF auto-EncodeR

## A   Training details of BRDF auto-encoder

For training our latent BRDF auto-encoder, we fetch diffuse, specular and roughness images from the TwoShotBRDF dataset and concatenate them along the channel dimension to form 7-dim BRDF material samples. Considering the BRDF material stored in each pixel is actually independent, we leverage an MLP network architecture to express our encoder and decoder, where the encoder contains 3 MLP layers and the decoder contains 4 MLP layers. During training, our encoder will consume a 7-dim BRDF parameter and embed it into 4-dim latent code, which is then decoded to a 7-dim reconstructed BRDF material via the decoder. For calculating the loss, we only consider the valid areas of each training sample, which are all indicated with a corresponding binary mask. Our BRDF auto-encoder is trained on an NVIDIA A100 GPU, which roughly takes $4.5$ hours to converge.

The training curves are illustrated in figure 1. After the training, we will embed all the training samples into latent space and calculate their mean $\mu$ and deviation $\sigma$ (Bi et al., 2020). While predicting the material, our material network $\Gamma$ will output a BRDF latent code $z$, which is then scaled by the deviation $\sigma$ and offset by the mean $\mu$ following:

$$z_{\boldsymbol{x}} = \Gamma(\beta(\boldsymbol{x}); \gamma) * \boldsymbol{\sigma} + \boldsymbol{\mu}. \tag{1}$$

This latent code $z$ will be decoded to BRDF material via our pretrained decoder $\mathcal{D}$.

## B   Additional results

In figure 2, additional material editing results on mug and chair cases are illustrated. It's noteworthy that we randomly select a chair object from the ShapeNet (Chang et al., 2015) dataset and obtain the corresponding 3D segmentation masks for free. As can be observed, we can edit its three independent parts alternately, across materials of gold, wood, and silver. Besides, more relighting results are illustrated in figure 3. All relit objects under different illuminations are natural and photorealistic. Moreover, we show the ablation study on the material smooth regularization weights in figure 4 and figure 5. From the comparisons, we can determine to set the regularization weights $\lambda_d$, $\lambda_r$, and $\lambda_m$ to 1e8, 10, and 100, respectively. We also test the material smoothness regularization proposed in prior work Munkberg et al. (2022) on the hammer case. As shown in figure 7, we empirically find our proposed regularization is more effective in separating the generated materials from the illuminations. Moreover, we include some implausible 3D objects generated with our method in figure 8. As recommended, we conduct several inverse rendering algorithms on the 3D objects from SOTA text-to-3D methods like Magic3D Lin et al. (2023) in figure 9. We also demonstrate the limitations shown in figure 6 stem from the bias in the text-to-image diffusion model in figure 10 and this is a particular limitation with statues by providing an extra case of wooden tiger statue 11. Furthermore, for the user study result, apart from the mean and standard deviation values presented in the main paper, we additionally include the Pearson correlation coefficient (PCC) between two different dimensions in table 1, the T-test results in table 2, as well as the interface of our user study in figure 12.

## C    LIMITATIONS

Our model sometimes still predicts diffuse materials entangled with unsatisfactory shadows. As shown in figure 6, there exist some weird shadows under the generated statue heads. We attribute this problem to the training data bias of stable diffusion models, which is not our focus here and is expected to be solved in future research. Besides, our model relies on 3D segmentation masks to determine different parts of the object to be painted. For those complex 3D objects, the corresponding 3D masks are extremely hard to obtain, even with the help of segment-anything model (Kirillov et al., 2023). Moreover, 3D assets generated by our current method still lack diversity and we are interested in circumventing it with promising Variational Score Distillation (VSD) loss proposed in Wang et al. (2023).

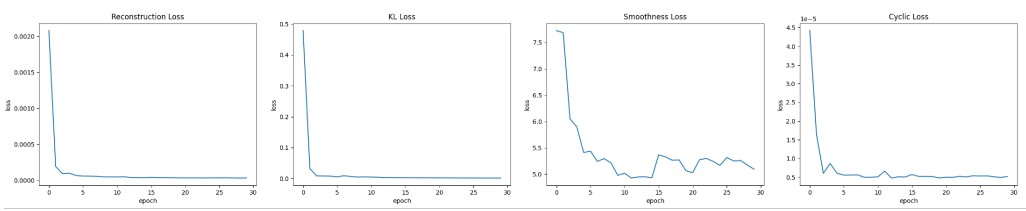

Figure 1: Training curves of our latent BRDF auto-encoder, where loss balancing coefficients $\lambda_{\mathrm{KL}}$, $\lambda_s$ and $\lambda_c$ are set to 2e-3, 1.0, and 1e-6, respectively.

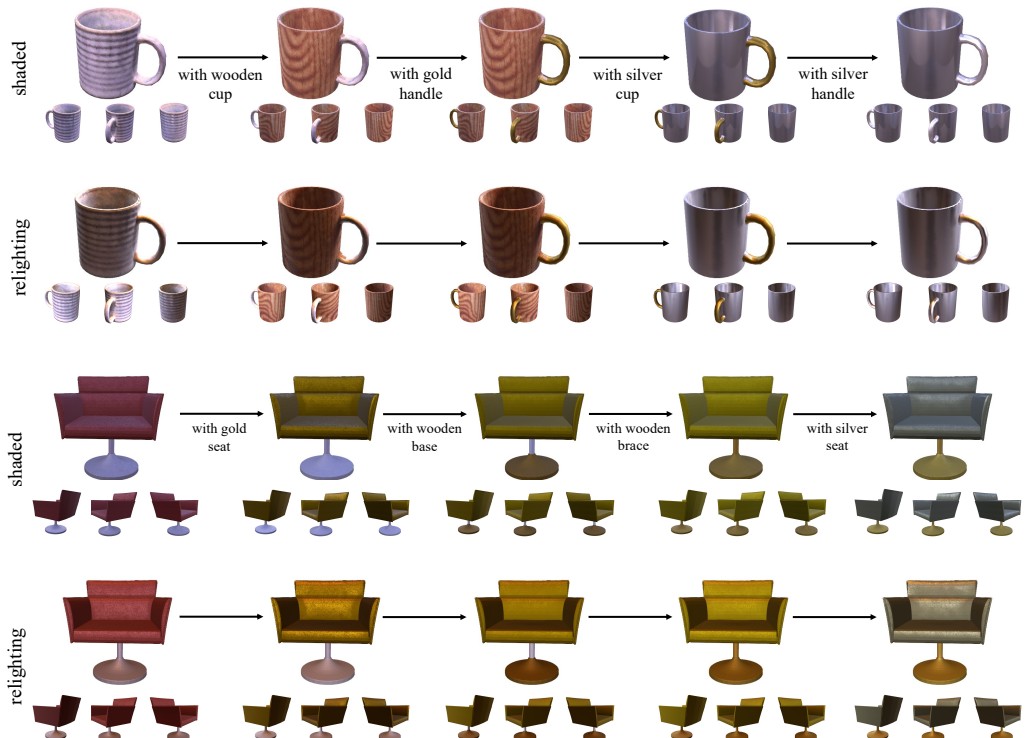

Figure 2: Additional material editing results. In the upper case, we start from the text prompt of "A ceramic mug", and alternately edit its cup and handle to wooden cup, gold handle, silver cup, and silver handle, respectively, under the guidance of corresponding text prompts. In the lower case, we demonstrate editing an object selected from the ShapeNet (Chang et al., 2015) dataset comprising three different parts.

Table 1: We calculate the Pearson correlation coefficient (PCC) between two different dimensions with all the collected data.

| Dimensions | Realism | Details | Disentanglement |
|---|---|---|---|
| Alignment | 0.149 | 0.267 | 0.048 |
| Realism | - | 0.674 | -0.125 |
| Details | - | - | -0.081 |

## REFERENCES

Sai Bi, Zexiang Xu, Pratul Srinivasan, Ben Mildenhall, Kalyan Sunkavalli, Miloš Hašan, Yannick Hold-Geoffroy, David Kriegman, and Ravi Ramamoorthi. Neural reflectance fields for appearance

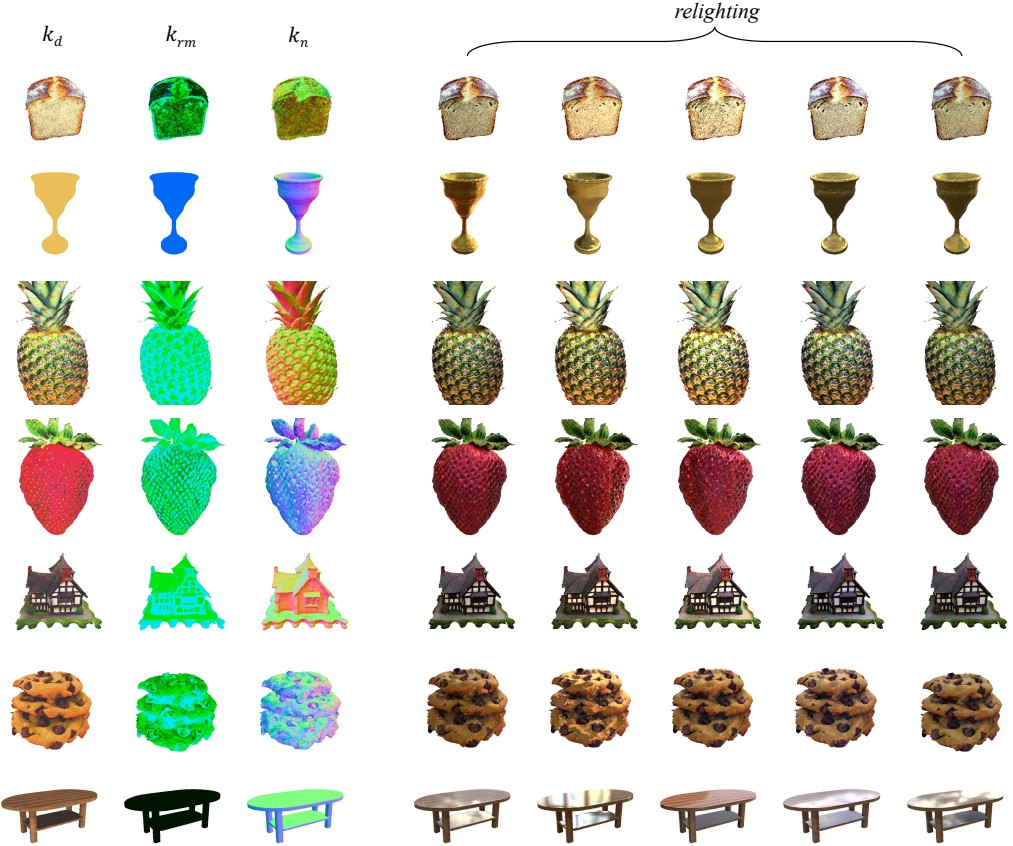

Figure 3: Additional relighting results. We put the generated materials on the left side, and list the relit images under 5 different environment maps. The table in the last row is borrowed from ShapeNet (Chang et al., 2015) and shows excellent relighting performance.

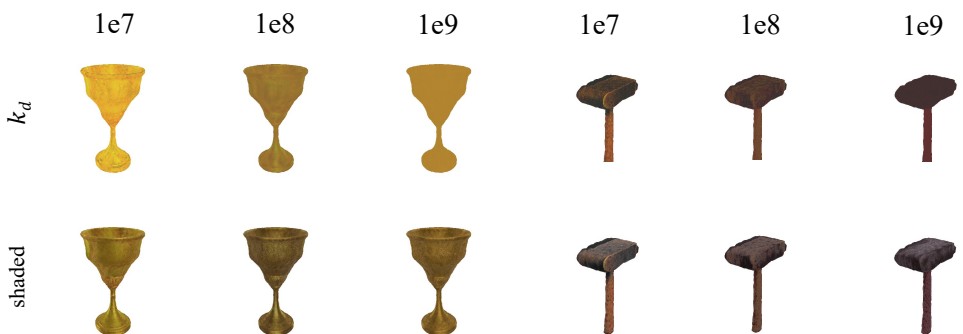

Figure 4: Ablation study on regularization weight of diffuse material $\lambda_d$. As can be observed here, the weight of 1e7 will make the highlights bake into the diffuse terms, and the weight of 1e9 leads to over-smooth rendering results. Our model achieves the best with the weight of 1e8.

acquisition. *arXiv preprint arXiv:2008.03824*, 2020.

Mark Boss, Varun Jampani, Raphael Braun, Ce Liu, Jonathan Barron, and Hendrik Lensch. Neural-pil: Neural pre-integrated lighting for reflectance decomposition. *Advances in Neural Information Processing Systems*, 34:10691–10704, 2021.

| | 1e1 | 1e2 | 1e3 | 1e0 | 1e1 | 1e2 |

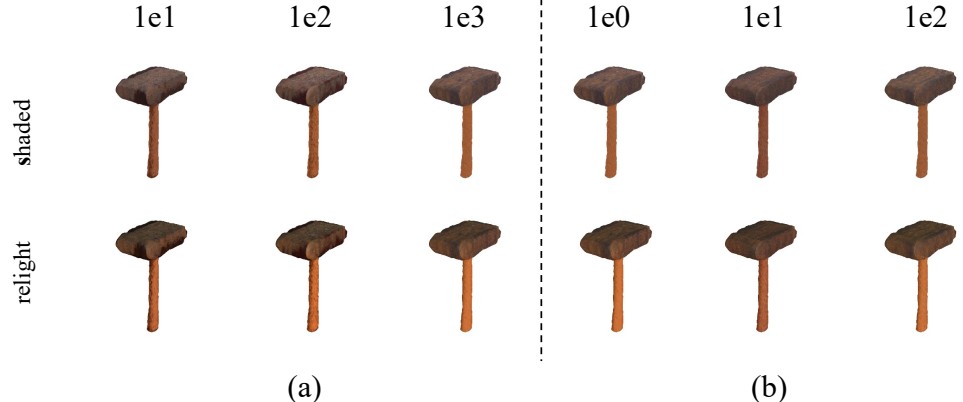

(a)  (b)

Figure 5: Ablation study on regularization weight of roughness $\lambda_r$ and metalness $\lambda_m$. On the left side, we show the ablation study on $\lambda_m$, where the weight of 100 leads to the sharpest edges on the hammer. The comparisons between different roughness regularization weights $\lambda_r$ are listed on the right side. We can observe rich variations in the texture of the hammer for the weight of 10.

| $k_d$ | shaded | relighting | $k_d$ | shaded | relighting |

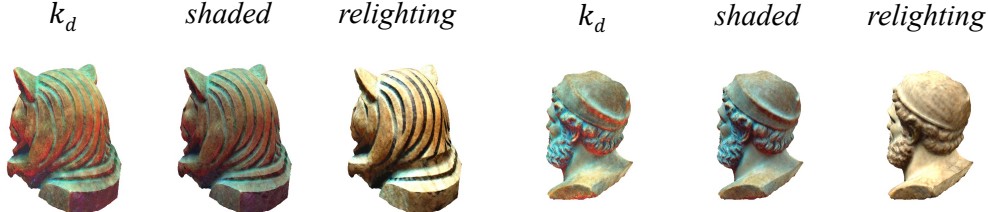

Figure 6: Limitations. Our model sometimes still struggles to disentangle diffuse materials from environmental shadows, especially in the statue cases.

Table 2: We conduct the one-sample T-test to validate the significance of our method's superiority over baselines in three dimensions, 'Realism', 'Details', and 'Disentanglement'. We report the p-value here and demonstrate our method outperforms baselines, especially in the 'Disentanglement' aspect. Only the significance level in the 'Detail' dimension is not enough ($0.58 > 0.05$) for the comparison between our method and the strong baseline Fantasia3D.

| p-value | Realism | Details | Disentanglement |
|---|---|---|---|
| Our $\succ$ DreamFusion | $1.20 \times 10^{-25}$ | $7.49 \times 10^{-47}$ | $2.36 \times 10^{-10}$ |
| Our $\succ$ Magic3D | $1.12 \times 10^{-13}$ | $1.35 \times 10^{-17}$ | $2.02 \times 10^{-19}$ |
| Our $\succ$ Fantasia3D | $4.22 \times 10^{-3}$ | 0.58 | $6.54 \times 10^{-27}$ |

Angel X. Chang, Thomas Funkhouser, Leonidas Guibas, Pat Hanrahan, Qixing Huang, Zimo Li, Silvio Savarese, Manolis Savva, Shuran Song, Hao Su, Jianxiong Xiao, Li Yi, and Fisher Yu. ShapeNet: An Information-Rich 3D Model Repository. Technical Report arXiv:1512.03012 [cs.GR], Stanford University — Princeton University — Toyota Technological Institute at Chicago, 2015.

Haian Jin, Isabella Liu, Peijia Xu, Xiaoshuai Zhang, Songfang Han, Sai Bi, Xiaowei Zhou, Zexiang Xu, and Hao Su. Tensoir: Tensorial inverse rendering. In *Proceedings of the IEEE/CVF Conference on Computer Vision and Pattern Recognition*, pp. 165–174, 2023.

Alexander Kirillov, Eric Mintun, Nikhila Ravi, Hanzi Mao, Chloe Rolland, Laura Gustafson, Tete Xiao, Spencer Whitehead, Alexander C. Berg, Wan-Yen Lo, Piotr Dollár, and Ross Girshick. Segment anything. *arXiv:2304.02643*, 2023.

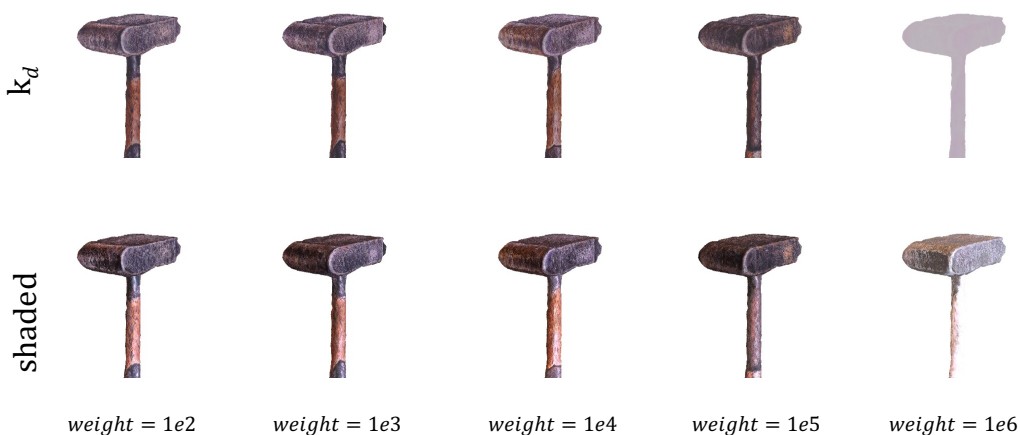

$k_d$

shaded

*weight = 1e2*  *weight = 1e3*  *weight = 1e4*  *weight = 1e5*  *weight = 1e6*

Figure 7: Validation of material smooth regularization. While loss weight is low, this regularizer can't prevent the diffuse map from entangling with the environment maps. However, a large loss weight will lead to weird materials and shaded images.

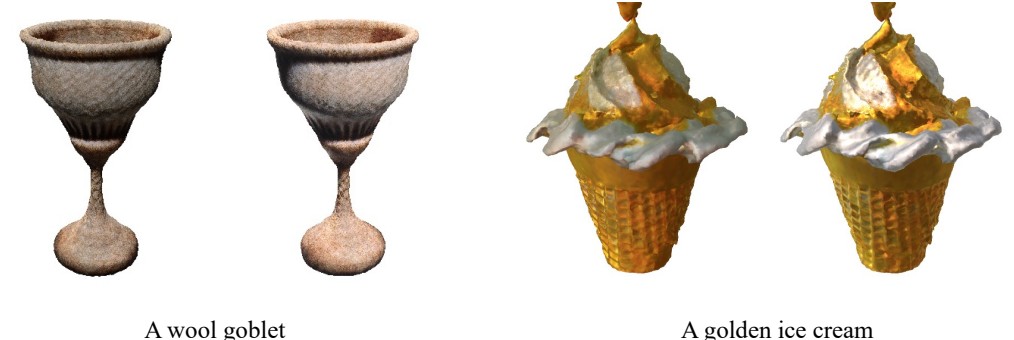

A wool goblet                                     A golden ice cream

Figure 8: Some generated implausible 3D objects. The shaded images under different illuminations are reasonable and aligned with the text prompt.

Chen-Hsuan Lin, Jun Gao, Luming Tang, Towaki Takikawa, Xiaohui Zeng, Xun Huang, Karsten Kreis, Sanja Fidler, Ming-Yu Liu, and Tsung-Yi Lin. Magic3d: High-resolution text-to-3d content creation. In *Proceedings of the IEEE/CVF Conference on Computer Vision and Pattern Recognition*, pp. 300–309, 2023.

Jacob Munkberg, Jon Hasselgren, Tianchang Shen, Jun Gao, Wenzheng Chen, Alex Evans, Thomas Müller, and Sanja Fidler. Extracting triangular 3D models, materials, and lighting from images. In *Proceedings of the IEEE/CVF Conference on Computer Vision and Pattern Recognition (CVPR)*, pp. 8280–8290, 2022.

Zhengyi Wang, Cheng Lu, Yikai Wang, Fan Bao, Chongxuan Li, Hang Su, and Jun Zhu. Prolific-dreamer: High-fidelity and diverse text-to-3d generation with variational score distillation. *arXiv preprint arXiv:2305.16213*, 2023.

Yuanqing Zhang, Jiaming Sun, Xingyi He, Huan Fu, Rongfei Jia, and Xiaowei Zhou. Modeling indirect illumination for inverse rendering. In *Proceedings of the IEEE/CVF Conference on Computer Vision and Pattern Recognition*, pp. 18643–18652, 2022.

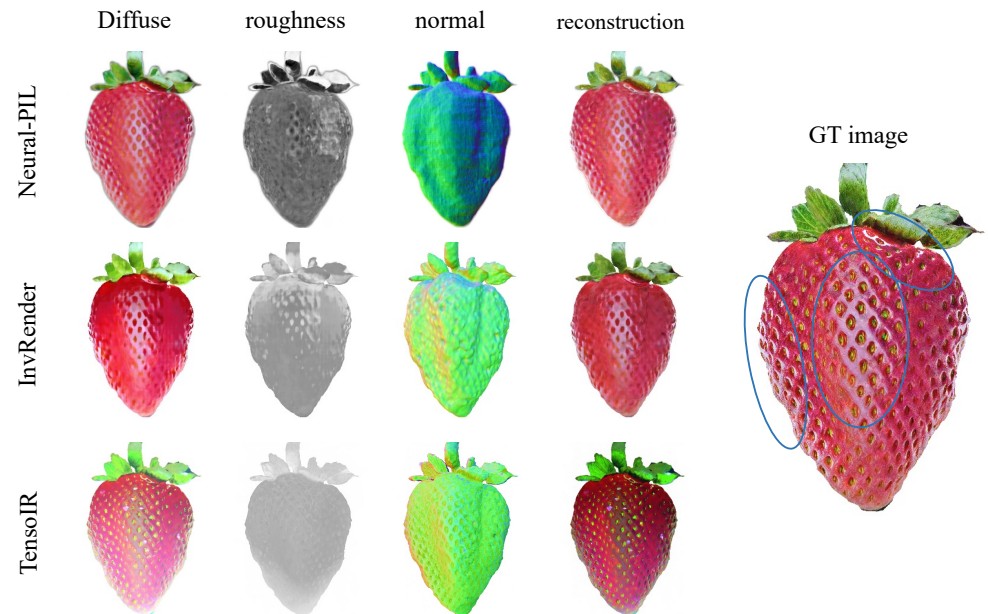

Figure 9: The rendered image from text-to-3D generation typically contains several bright lights (the circled areas), which often does not appear under normal circumstances. We have tested three different inverse rendering algorithms (Neural-PIL Boss et al. (2021), InvRender Zhang et al. (2022) and TensoIR Jin et al. (2023)) on the generated ripe strawberry. However, these methods cannot faithfully reconstruct the image and disentangle the diffuse map from the environmental lights.

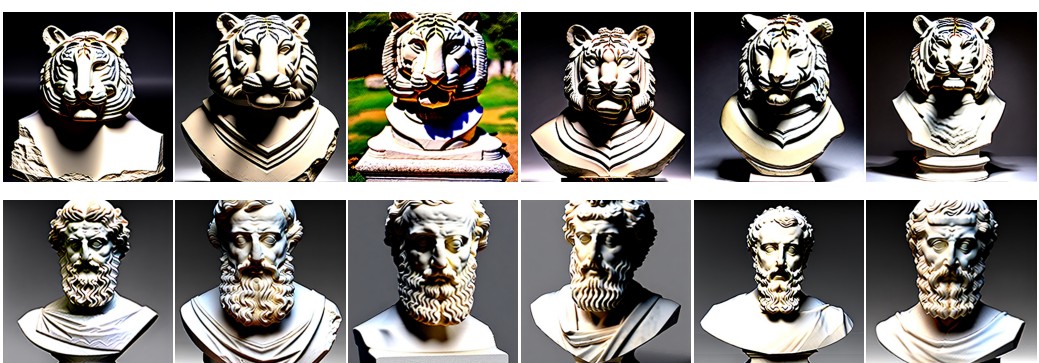

Figure 10: We show several statue images generated by the stable diffusion model with different seeds, which demonstrates that the text-to-image diffusion model is prone to generate statue images will clear shadows, especially the area under the head.

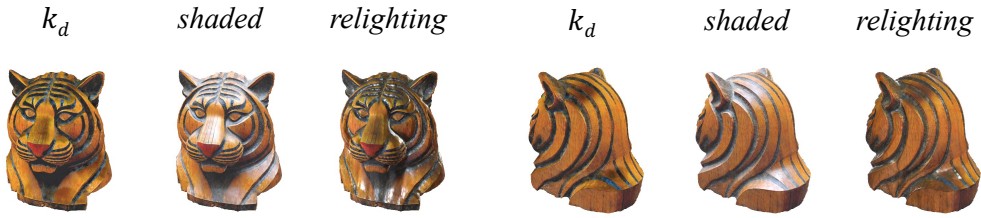

Figure 11: We generate a wooden tiger statue where the diffuse map still contains some unsatisfactory shadows.

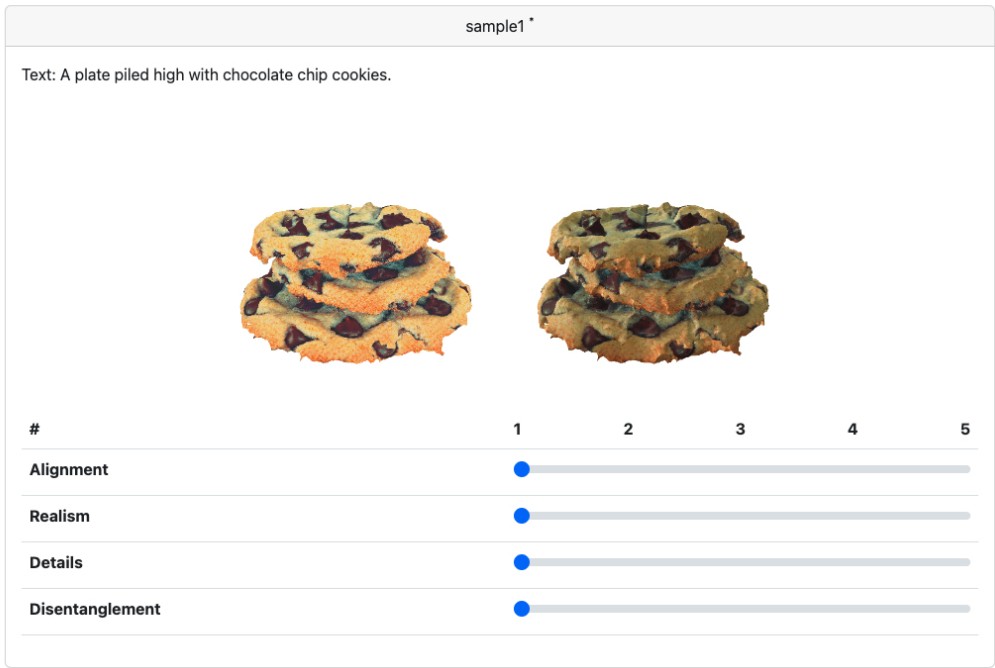

Figure 12: We present the interface of our user study.