# OpenReview forum: "MATLABER: Material-Aware Text-to-3D via LAtent BRDF auto-EncodeR"
_ICLR.cc/2024/Conference — Submitted to ICLR 2024_

### Official Review · Reviewer_A1cD · 2023-10-17

**Soundness:** 3 good
**Presentation:** 3 good
**Contribution:** 2 fair
**Rating:** 6
**Confidence:** 4

**Summary:**

This paper introduces a novel method for material-aware text-to-3D object generation. Previous work on this topic are limited in the sense that shading effects are baked into the material reflectance properties, thus not allowing for important applications like relighting or material interpolation, and reducing their quality. This paper introduces a BRDF autoencoder which allows text-to-3D generative models to return full BRDF parameters, by leveraging a VAE trained on a large dataset of BRDFs. This autoencoder, combined with a pre-trained text-to-image model and a radiance field representation, allows for the generation of 3D objects from text prompts with disentangled material properties, including surface albedo, roughness or normals. The paper evaluates their results with ablation studies and a user study, in which they compare their work with previous text-to-3D models across a variety of metrics.

**Strengths:**

- This paper introduces an interesting and sound solution to a key limitation of text-to-3D generative models. Disentangling geometry (or shading) from reflectance remained a challenge for such generative models and this work introduces a valuable solution to this problem.
- The method introduced in this paper is sound, and combines ideas from classical computer graphics and more modern neural rendering and generative model techniques in an interesting way, which may be valuable for many downstream applications and other problems.
- The method is evaluated on a large variety of materials and objects, and the results show improved quality with respect to baselines.
- The qualitative analysis is sound, and the user study, albeit limited, provides insights on user preferences for different works.
- The paper is very well structured and mostly well written, making it easy to follow.
- The material interpolation and edition methods are very interesting and the results are impressive.
- The ideas for improving the VAE results and the semantic-aware material regularization are sound and may benefit future work on different topics.
- The supplementary material provides valuable insights on the quality of the results and the impact of different individual components.

**Weaknesses:**

- I believe the paper could benefit from a better motivation. In particular, it is not clear why separating shading from reflectance is difficult and which approaches exists for this problem. Further, the lambertian assumption and its limitations require better explanation, to help the reader understand what are the challenges that this paper addresses and why the solution is valuable. I suggest looking into and referring the reader to "A Survey on Intrinsic Images: Delving Deep Into Lambert and Beyond, Garces et al. IJCV 2022) for a contextualization of this problem.
- The related work analysis is somewhat limited. First, I think this paper requires a more in-depth analysis of radiance field representations (At least Gaussian Splatting should be mentioned). Second, more recent work on generative models for material estimation should be included (ControlMat, UMat, SurfaceNet, etc.). Importantly, work on BRDF compression should also be analyzed (eg the work by Gilles Rainer et al. on neural BRDFs and BTF compression). Recent work on text-to-3D is missing (SMPLitex, HumanNorm), although these may be concurrent and thus not applicable to this submission. Finally, I think that this paper is also missing an analysis of illumination representations, as the authors only test basic environment maps, but other approximations (point lights, spherical gaussian, neural illumination approximations, spherical harmonics, etc) also exist and I believe should be mentioned.
- I am doubtful about the soundness of some parts of the method, in particular Sections 3.2 and 3.3. (See the Questions section).
- I have several concerns regarding the validation of this method, particularly in the ablation and the user study (See the Questions section).
- There are important details missing, particularly in terms of computational cost.
- The results are sometimes of a low quality (the geometries are sometimes very coarse and not sharp). While this is a limitation shared with previous work, I believe that it should at least be mentioned in the paper.
- There is no analysis of limitations or suggestions for future work. I believe these should be included.
- Implementation details are not enough for reproducibility.

**Questions:**

- How much of the capabilities of this method are linked with the radiance field representation that was chosen? That is, why was MIP-NeRF chosen and what would happen if other model was selected instead?
- Why was Cook-Torrance chosen as the material model? What would happen if a more complex or a simpler model was used instead? This material model, in the form explained in the paper, does not model anisotropy, among other reflectance properties. This limits the generality of the materials and objects that can be generated with them.
- Why was the TwoShotBRDF dataset used? There are plenty of other datasets of SVBRDFs available, of higher resolutions and with a different diversity of material classes. I am wondering how many of the limitations of the method (eg it struggles with metallic objects) are due to the dataset choice.
- How were the hyperparameters of the different losses selected? What are their impact?
- In section 4, could the authors provide an in-depth analysis of the computational cost of each part of the method? I think it would be interesting to see timings, memory usage and FLOPs.
- The authors mention that "we initialize DMTet with either a 3D ellipsoid, a 3D cylinder, ...". How is this selected? Is it automatic?
- What are the demographics of the user study? Are they instructed in any meaningful way? Can the authors provide a more detailed description of the test that each participant undertook? I am not convinced that it is fair to ask random people to measure the "disentanglement" of a text-to-3D generative model, as this is very hard to evaluate even for experts or automatic metrics.
- How correlated are the generated objects with their materials? For example, if prompted for a "goblet", does it always generate metallic objects or are more variations allowed? How well does it generate implausible objects (eg "a wool goblet" or a "ceramic pizza", etc)?
- How does this method handle fibrous objects (eg a fleece fabric or a knitted teddy bear)?
- I suggest the authors include the illumination used in every render, particularly when showing results of relighting. It is unclear which type of illumination was used. Given the images that are shown, my guess is that very diffuse illumination was used to render the objects, which makes me wonder how well these 3D objects look on more directional illumination.
- Could the material MLP be used to generate different set of BRDF parameters from different material models (eg Phong, complex Cook-Torrance, etc.) from the same latent space? If so, how would it impact the final results?
- Transparency and surface scattering is very important in many real world materials and basic BRDFs cannot model such behaviours. How would the authors extend this work in order to generate full BSDFs? NeRF explicitly models alpha in their MLPs. Could this be combined with the Cook-Torrance BRDF in any way so as to allow for modelling these more complex effects?



Writing improvement suggestions:
- Page 1, Paragraph 2: "The neural network ... has no sufficient motivation". This sentence is anthropomorphizing the neural network and could be written in a different, more technical, way.
- Page 1, Paragraph 3: "There exist$\textbf{s}$",
- Page 3, Section 2.3: "Unluckily" could be changed to "Unfortunately", which sounds better in my opinion.
- Page 7: "he/she" --> "they"

**Details Of Ethics Concerns:**

The authors have mentioned that "I certify that there is no URL (e.g., github page) that could be used to find authors' identity." on their submission. However, I came across this URL which is not anonymized: https://github.com/SheldonTsui/Matlaber . I am not sure this complies with ICLR code of conduct and I would ask the ACs to look into this. My review has not been influenced by this in any way.

---

> ### Author Response · Authors · 2023-11-14
> **Your insightful questions will help to improve our paper**
>
> **Clarification**
>
> It's worth noting that our main target in this paper is to **introduce a material prior** to text-to-3D to generate 3D objects with proper materials, rather than studying the optimal representation in text-to-3D. As recognized by reviewer Kg7h, our work is **the first work** to introduce such a prior in this line of research.
>
>
> **Q1: This paper could benefit from a better motivation.**
>
> A: Thanks for your suggestion, and we will refine our manuscript based on your comments. It's worth noting that the mentioned IJCV paper is more related to inverse rendering (a reconstruction task), which is conceptually different from text-to-3D (a generation task). In text-to-3D, only a text input provides loose semantic constraints about the shading and the reflectance, leading to a lot of ambiguities and uncertainties that simply applying inverse rendering techniques can not resolve.
>
>
> **Q2: Why was Mip-NeRF chosen? More related work should be included.**
>
> A: As discussed in the **Clarification**, our focus of this paper is on the material side rather than the representation side, thus discussing works on 3D representations in the related work and comparing different 3D representations may deviate from our focus. As mentioned in Section 3.4, we followed previous studies and used **DMTet, not Mip-NeRF**, as our 3D representation. Finally, we will revise our manuscript to include possible related work on material estimation, BRDF compression, and illumination representations.
>
>
> **Q3: Why was the TwoShotBRDF dataset used? Why was the TwoShotBRDF dataset used? How to handle fibrous objects?**
>
> A: As mentioned, our work is the first one to introduce material prior to the task of text-to-3D. Being the first attempt, we choose Cook-Torrance as the material model and TwoShotBRDF as the source of our material prior, as they cover a large portion of possible materials, and we believe they are sufficient to verify the effectiveness and importance of involving such a material prior to text-to-3D. Our experiments also demonstrated this. Specifically, the Cook-Torrance model is a widely used material model in many commercial softwares such as UE4 and academic works. TwoShotBRDF is a large-scale BRDF dataset containing most of the commonly observed materials. In terms of fibrous objects, both its geometry and material are too challenging for current text-to-3D. Involving other material models, as well as adding more datasets to enhance the material prior is worth exploring, which is left as future work. Same as the handlement fibrous objects. We will clarify this in the revision.
>
>
> **Q4: Details about hyperparameters, computational cost, and initial geometry.**
>
> A: We will clarify this in the revision. Also, code and model to reproduce our results will be released. For hyperparameters of different losses, we borrow some part of the default setting of Neural-PIL and fine-tune them to ensure four losses decrease simultaneously.  For the computational cost, the optimization of each object will take around 40 minutes (20 minutes for geometry and another 20 minutes for appearance) on 4 A100 GPUs, and about 30G memory is used in each GPU. Finally, we follow standard practice in text-to-3D (e.g. Fantasia3D) to select initial geometry, which is ruled by the configuration file of each 3D object to be optimized. This part is not investigated in-depth as it is not our main focus.
>
>
> **Q5: Details of user study.**
>
> A: For the dimension of "disentanglement", we will present the shaded images of generated 3D objects and the corresponding diffuse map to the participants, and ask them whether the diffuse map contains weird and unexpected bright lights or shadows.
>
>
> **Q6: How well does this method generate implausible objects?**
>
> A:  Thanks for your suggestion. Our method is capable of generating implausible objects like "a wool goblet" and "a golden ice cream". We update our manuscript and include these results in Appendix Figure 8 (supplementary). More cases will be included later.
>
>
> **Q7: The illumination used in every render should be included.**
>
> A: Thanks for your suggestion, and we will include the illumination in the revision.
>
>
> **Q8: Could the material MLP be used to generate different sets of BRDF parameters from different material models from the same latent space?**
>
> A: It is an interesting direction to explore in the future. It's possible to learn different latent spaces or a shared latent space for different material models, and subsequently use a set of MLPs to generate multiple sets of BRDF parameters.
>
>
> **Q9: Limitations.**
>
> A: The discussion of limitations is included in the supplementary.

---

> > ### Comment · Reviewer_A1cD · 2023-11-17
> > **Rebuttal response**
> >
> > Thank you for your detailed response to my review, they were helpful responses and I hope these clarifications make their way to the final paper.
> >
> > I am also glad to read the code will be relased.
> >
> > I strongly suggest to include some of the limitations on the main paper. I saw the figure about the limitations on generating marble-like materials (FIG 6 in the supplementary). Why did this happen? Is it a limitation of the training dataset or just a particular limitation with statues?

---

> > > ### Author Response · Authors · 2023-11-22
> > > **Thank you for your response**
> > >
> > > **Q10: Why did artifacts happen with marble-like materials? Is it a limitation of the training dataset or just a particular limitation with statues?**
> > >
> > > A: We hypothesize these artifacts are more related to the bias of the T2I diffusion model used in the text-to-3D pipeline. Specifically, T2I diffusion models tend to generate images with shadows, especially statue images, due to their training data bias and their objective of generating realistic images (Appendix Figure 10). The only objective of the text-to-3D pipeline is to push all the rendered images to the high probability density regions given by the T2I models, and the SDS loss uses the T2I diffusion model to assess the quality of rendered images. Subsequently, the bias of the T2I diffusion model leaks into 3D content generated by the text-to-3D pipeline, causing artifacts observed in marble-like materials. Apart from the marble material, we also try the wooden material on the tiger statue in Appendix Figure 11 (supplementary), where the diffuse map still contains some unexpected shadows. By the way, we will move the limitations from the supplementary to the main paper.

---

### Official Review · Reviewer_Kg7h · 2023-10-18

**Soundness:** 4 excellent
**Presentation:** 4 excellent
**Contribution:** 3 good
**Rating:** 5
**Confidence:** 5

**Summary:**

This paper proposes a novel method to generate 3D assets with more disentangled reflectance maps by exploiting 2D diffusion priors and BRDF priors. To further improve disentanglement, a novel loss function is adopted to encourage piece-wise constant material. To me, the paper is well-written and easy to follow.

**Strengths:**

* The paper works on an important problem, i.e. generating 3D assets with reflectance maps.
* The paper improves previous work's results by incorporating more priors.
* The paper performs a user study to demonstrate its advantage over the previous works.
* The application demonstrated in the paper is interesting, including material editing and interpolating.

**Weaknesses:**

(1) To me, the techniqical contribution is limited.
* Leveraging a trained BRDF prior to regularize the inverse rendering algorithm is a common way in the literature. As the author discussed in the related works, Neural-PIL and NeRFactor do very similar things. Other works also introduce a low rank prior to the spatially varying BRDF[1][2].
* I know that the paper is the first to introduce this prior in 3D AIGC. Are there more challenges to using the BRDF prior in the 3D AIGC pipeline?

(2) The semantic-aware material regularization is not well evaluated in my eyes.
* Many previous works have proposed techniques to regularize the material values, e.g., Munkberg et al., 2022. To me, if the author claimed L_mat is their main contribution, more comparisons to previous techniques are expected. However, the paper only compares their method to the w/o L_mat baseline.

[1] Neural reflectance for shape recovery with shadow handling, CVPR2022

[2] NeuFace: Realistic 3D Neural Face Rendering from Multi-view Images, CVPR2023

**Questions:**

I find that the paper randomly samples the environment maps from a pool of collections and randomly rotates the map during training. I think such a multi-light setup can reduce the ambiguity in the inverse rendering process a lot. However, as shown in Figure 8, the reconstructed albedo is disentangled without the BRDF prior and the L_mat. Can the author provide some insights on these unexpected results?

---

> ### Author Response · Authors · 2023-11-14
> **Thank you for your recognition and feedback**
>
> **Q1: Difference between Text-to-3D and Inverse Rendering**
>
> Thanks for recognizing that our work is the first to introduce the material prior to 3D AIGC, particularly text-to-3D. We argue that **inverse rendering and text-to-3d are two conceptually different tasks**. They face different challenges as **theirs are respectively reconstruction and generation tasks**. Specifically, inverse rendering aims at **reconstructing the correct materials** given single or multi-view images, where these multi-view images provide strong constraints to eliminate any ambiguity.  On the contrary, our text-to-3D method aims at **generating proper materials** when generating a 3D object given only a text input. Such a text input only provides loose semantic constraints on the possible materials, leading to lots of ambiguity and a much larger set of valid materials.
>
>
> **Q2: The semantic-aware material regularization is not well evaluated.**
>
> A: In the last sentence of the method section, we state that our proposed regularization $\mathcal{L}_\text{mat}$ is more effective than the material smoothness regularization proposed in prior works (Zhang et al 2021, Munkberg et al, 2022). As requested, we include a comparison between different regularizations in Appendix Figure 7 (supplementary). More cases will be updated in a few days. From the case we can see, that text-to-3D requires not only two **geometrically close points** to have similar materials but also two **semantically close points** to have similar materials. Therefore previous material smoothness regularization is not sufficient. This is also an important challenge of text-to-3D that differentiates it from inverse rendering.
>
>
> **Q3: Under multi-light setup, why is the albedo still engtangled with the environment map?**
>
> A: This is exactly a specific example of challenges mentioned in Q1 that differentiate text-to-3D from inverse rendering. The task of text-to-3D only has a text input that provides loose semantic constraints to regularize generated materials via the SDS loss. The SDS essentially maximizes the similarity between the text input and images rendered from the target 3D content, and we empirically found some tiny bright lights and shadows in the rendered images can easily fool the SDS loss for a higher similarity. To summarize, although our material prior provides more information to help generate proper materials, and experiments demonstrate its effectiveness, the task of text-to-3D is very challenging and some ambiguities need further investigation to resolve.

---

> ### Comment · Reviewer_Kg7h · 2023-11-15
>
> Firstly I want to thank the authors for the reply. According to the reply, I raise a further question.
>
> Can the author comment on the expected comparison results of the following methods:
> 1. generate material and geometry jointly, like the paper's method and the Fantasia3D work.
> 1. a 2-stage method that combines DreamFusion (or recent SOTA 3D generation work, e.g. Magic3D, DreamCraft3D, DreamGaussian) to some inverse rendering method (e.g. Nerfactor, TensoIR). In this method, in the first stage, we generate 3D objects aligned with the text, and in the second stage, we disentangle the material from the baked texture.
>
> I agree that text-to-3d is a generation task. But I wonder if material estimation in the paper's task is also a generation task. If yes, what is the benefit of treating it as a generation task instead of a reconstruction task?

---

> > ### Author Response · Authors · 2023-11-22
> > **Thanks for your suggestions**
> >
> > **Q4: Is material estimation in the paper's task also a generation task? If yes, what is the benefit of treating it as a generation task instead of a reconstruction task?**
> >
> > A: There is no material estimation in the vanilla text-to-3D pipeline, which just tries to predict densities and **RGB values** that can render text-aligned images. Some attempts such as Fantasia3D embed material estimation in text-to-3D, but face significant challenges as the condition (a text input) is insufficient to resolve the ambiguities and uncertainties. Our proposed Matlaber estimates materials in text-to-3D in a generative way, where it at first learns a distribution of valid materials, followed by **choosing** text-aligned valid materials from the distribution. The benefits can be observed from the suggested comparison. We thank the reviewer for this suggestion. We update our manuscript and include additional results in Appendix Figure 9 (supplementary), where one can see current SOTA text-to-3D methods are prone to hallucinate multiple bright lights in the rendered images. Specifically, we leverage Magic3D to synthesize a ripe strawberry and implement several inverse rendering methods (Neural-PIL, InvRender, TensoIR) on multiview rendered images. As can be observed, these methods still struggle to disentangle the diffuse map from the bright lights.

---

### Official Review · Reviewer_VHfX · 2023-10-31

**Soundness:** 2 fair
**Presentation:** 2 fair
**Contribution:** 2 fair
**Rating:** 3
**Confidence:** 3

**Summary:**

In this manuscript, the authors investigated a framework to generate material appearance in a text-to-3D latent diffusion model. They utilized a latent BRDF auto-encoder and compared their results with existing models.

**Strengths:**

Estimating material properties is often overlooked in image generation. I agree that including them is essential for future image generation. However, I'm not convinced with the authors' model in terms of the following points.

**Weaknesses:**

The BRDF is one of the descriptions of physical material properties. Many natural objects include more than reflection, like absorption or sub-surface scattering. Adding the constraint of BRDF in their model makes the material appearance of output images narrower than other methods, like Fantasia3D. For example, the ice cream in Figure 3 by Fantasia3D looks translucent, but the authors' output lacks such a translucent material appearance, which is critical for foods.

In addition, the diffuse component of gold in Figure 1 is weird. The ground truth of yellow components for gold materials comes from the specular reflection of metals, not from diffuse components. The model does not look to capture material properties.

For the user study, the authors did not conduct any statistical tests. They cannot conclude anything without them.

**Questions:**

The authors compare their method only with text-to-images. However, in particular, material editing in Figure 7 has a long history in the Computer Graphics community, and many methods have been developed. The text-to-image is not only the way to edit material appearance. The authors should also compare them with their editing.

---

> ### Author Response · Authors · 2023-11-14
> **Thanks for your valuable feedback**
>
> **Clarification of the Task**
>
> It's worth noting that our task is **conceptually different from material estimation in image generation**. As stated in the introduction, our task is text-to-3D, which **generates 3D objects given a text input**. And, the goal of this paper is to **introduce a material prior** so that generated 3D objects have more realistic and coherent materials, which, as recognized by reviewer Kg7h, **is the first work in this line of research**.
>
>
> **Q1: Why not consider absorption or sub-surface scattering, other than reflection?**
>
> A: Being the first work that explores a material prior to text-to-3D, we prioritize tackling reflections first as it covers a large portion of possible object materials, and currently is more tractable. We thus believe starting from BRDF is a good choice and sufficient for us to verify the effectiveness of such a material prior. For absorption, sub-surface scattering, and other cases, we leave them as future work where we can explore various potential solutions to enhance the material prior. For example, we could leverage hybrid material MLPs to model absorption or sub-surface scattering as well.
>
>
> **Q2: The diffuse component of gold in Figure 1 is weird.**
>
> A: For the diffuse component of gold in Figure 1, we follow the conventions in UE4 (Karis et al 2013), where the diffuse term $\mathbf{k}_d$ can’t be applied to the rendering equation directly. As mentioned in Section 3.2, we compute the specular term **for rendering** with $\mathbf{k}_s = (1-m) \cdot 0.04 + m \cdot \mathbf{k}_d$, and the diffuse term **for rendering** with $(1-m) \mathbf{k}_d$. Hence, for metal material whose metalness $m$ is close to 1, the yellow components for gold materials are indeed from the specular reflection. We will clarify it in the revision.
>
>
> **Q3: The user study without any statistical tests.**
>
> A: We followed previous works in text-to-3D to present our user study. Thanks for the suggestion, we will try to provide the statistical tests in a few days.
>
>
> **Q4: Compare with traditional material editing methods in the CG community.**
>
> A: As discussed, we are the first work that explores material prior to text-to-3D. Currently, it is non-trivial to apply traditional CG methods to text-to-3D for material editing, which contains only a text input without any UV mapping. Nevertheless, integrating traditional CG methods into text-to-3D is definitely a promising direction we shall explore in the future.

---

> ### Author Response · Authors · 2023-11-22
> **Looking Forward to Your Feedback**
>
> Thank you again for your valuable feedback, which we already provided detailed responses to. We hope our responses address your concerns. As the deadline for discussion is approaching, we'd like to ask if you have further comments. Looking forward to your reply :)

---

> > ### Comment · Reviewer_VHfX · 2023-11-22
> > **Responses to Authors**
> >
> > I understand the main purpose of this study. However, if the authors evaluate the material editing in their application, it should still be compared with other methods in computer graphics literature.
> >
> > In addition, the authors have yet to share the statistical evaluation of user tests after their responses, although they commented that they planned to share it in a few days. I am not convinced in terms of the evaluation of their methods.

---

> > > ### Author Response · Authors · 2023-11-23
> > > **Thanks for your responses**
> > >
> > > **Q5: However, if the authors evaluate the material editing in their application, it should still be compared with other methods in computer graphics literature.**
> > >
> > > A: As far as we know, material editing in computer graphics literature relies on a UV map. As discussed in our previous response, the UV map is not included in the current text-to-3D pipeline, making it infeasible to conduct a fair and reasonable comparison.
> > >
> > > **Q6: Statistical evaluation of user tests**
> > >
> > > A: Apart from the mean and standard deviation values presented in Table 1, we additionally conduct the T-test and calculate the Pearson correlation coefficient (PCC) between two different dimensions, and the results are included in Appendix Table 1&2, which indicate our user study can reflect the effectiveness of Matlaber. It's worth noting that we follow the standard evaluation protocol as in [1,2,3], which includes a user study and various ablative qualitative comparisons.  Besides user study, we also provide rich qualitative results in the main experiments (Fig.3, Fig.4, Fig.5, Fig.6) and appendix (Fig.3).
> > >
> > >
> > > References:
> > >
> > > [1] Magic3D: High-Resolution Text-to-3D Content Creation. Lin et al, CVPR 2023
> > >
> > > [2] Fantasia3D: Disentangling Geometry and Appearance for High-quality Text-to-3D Content Creation. Chen et al. ICCV 2023
> > >
> > > [3] ProlificDreamer: High-Fidelity and Diverse Text-to-3D Generation with Variational Score Distillation. Wang et al. NeurIPS 2023

---

> > > > ### Comment · Reviewer_VHfX · 2023-11-23
> > > > **Responses to Authors**
> > > >
> > > > I don't understand why the correlation analysis the authors did could address the user evaluation.
> > > > Additionally, presenting only p-values is insufficient for the t-test; the author should also provide t-values and degrees of freedom. Moreover, critically, the t-tests conducted by the authors should be corrected for multiple comparisons, such as Bonferroni correction. In this case, the authors performed t-tests 24 times, according to Table 2 of the main text. Finally, referencing other studies that do not employ statistical evaluations is not convincing to me, even if they are from top conferences. I am emphasizing the need for scientific evaluation in this paper, rather than relying on common sense within these communities.
> > > >
> > > > Besides, I understood that the authors don't plan to compare their methods with other methods I suggested. However, there should be many ways to transfer materials without UV maps.

---

> > > > > ### Author Response · Authors · 2023-11-23
> > > > > **Thanks for your responses**
> > > > >
> > > > > We thank the reviewer for further detailed suggestions on statistical tests on our user study. Due to the time limit of rebuttal, we cannot provide these tests in time. We will try to conduct these tests and include them in the revision. Besides, we will try to implement the approaches to transfer materials without UV maps and compare our method with them.

---

### Meta-Review · Area_Chair_5ie3 · 2023-12-12

**Metareview:**

The paper has received diverged reviews with one acceptance and two rejection recommendations. After considering the rebuttal, two reviewers still remain the recommendation for borderline rejection.

The main strength of this paper is that it introduces a material prior for text-to-3d generation tasks. The prior work modeling materials in text-to-3d generation like Fantasia3D does not use such prior. The paper argues that without the material prior, prior work fails to disentangle lighting and materials.

The main issue is on the evaluation of the proposed method. All reviewers expressed the same concern on the evaluation. Reviewer A1cD expressed the concern about the validity of user study because the disengagement of material and light is challenging even for experts. Reviewer Kg7h concerned with comparisons with existing works. More ablation studies are presented in the supplementary material and addressed some concerns. Reviewer VHfX asked for a rigorous user study for model quality.

In the post-rebuttal discussion,  reviewer A1cD commented that perceptual evaluation is a generally challenging topic. Reviewer VHfX’s comment on the missing rigorous user study might be too harsh a reason for rejecting the paper. Reviewer VHfX agrees and is willing to increase the rating to borderline rejection. Reviewer A1cD summarizes that they agree the evaluation and the comparison to existing works can be improved but the overall contribution is sufficient for publication. However, reviewer Kg7h and VHfX still hold their views on insufficient evaluation and comparisons after rebuttal.

After careful consideration of the review, rebuttal, and post-rebuttal discussion, AC concurred with the assessments from all reviewers. AC agreed that this paper has a sound idea to bring the material prior into 3D generation. However, the paper does not provide enough qualitative visualization (such as a variety of gendered assets with light information) and quantitative analysis to support the proposed method. There are still limitations of the proposed method such as light baking in certain text prompts. However, there is no rigorous analysis about when the proposed method and prior method would succeed or fail. Consequently, the AC recommends rejecting the paper and encourages authors to improve presentations for both qualitative and quantitative evaluation of the proposed method and conduct rigorous analysis to demonstrate the improvements over existing methods.

**Justification For Why Not Higher Score:**

While the paper proposes an interesting and sound idea to bring material priors into text-to-3d generation. The paper misses enough qualitative and quantitative analysis to demonstrate the improvements over existing methods and discuss the limitations.

**Justification For Why Not Lower Score:**

N/A

---

### Decision · Program_Chairs · 2024-01-16

Reject